# Fiber-Rich Barley Increases Butyric Acid-Producing Bacteria in the Human Gut Microbiota

**DOI:** 10.3390/metabo11080559

**Published:** 2021-08-22

**Authors:** Shohei Akagawa, Yuko Akagawa, Yoko Nakai, Mitsuru Yamagishi, Sohsaku Yamanouchi, Takahisa Kimata, Kazushige Chino, Taiga Tamiya, Masaki Hashiyada, Atsushi Akane, Shoji Tsuji, Kazunari Kaneko

**Affiliations:** 1Department of Pediatrics, Kansai Medical University, 2-5-1 Shinmachi, Hirakata 573-1010, Japan; akagawas@hirakata.kmu.ac.jp (S.A.); akagaway@hirakata.kmu.ac.jp (Y.A.); yokodayo.co@gmail.com (Y.N.); yamagism@hirakata.kmu.ac.jp (M.Y.); yamanous@hirakata.kmu.ac.jp (S.Y.); kimatat@hirakata.kmu.ac.jp (T.K.); tsujis@hirakata.kmu.ac.jp (S.T.); 2Healthcare New Business Division, TEIJIN Limited, 3-2-1 Kasumigaseki, Tokyo 100-8585, Japan; k.chino@teijin.co.jp; 3Bio Palette Co., Ltd., 1-1 Rokkodai-cho, Kobe 650-0047, Japan; tamiya.taiga@gmail.com; 4Department of Legal Medicine, Kansai Medical University, 2-5-1 Shinmachi, Hirakata 573-1010, Japan; hashiyam@hirakata.kmu.ac.jp (M.H.); akane@hirakata.kmu.ac.jp (A.A.)

**Keywords:** microbiota, butyric acid-producing bacteria, 16S rRNA gene sequencing, dietary fiber, prebiotics

## Abstract

Butyric acid produced in the intestine by butyric acid-producing bacteria (BAPB) is known to suppress excessive inflammatory response and may prevent chronic disease development. We evaluated whether fiber-rich barley intake increases BAPB in the gut and concomitantly butyric acid in feces. Eighteen healthy adults received granola containing functional barley (BARLEYmax^®^) once daily for four weeks. Fecal DNA before intake, after intake, and one month after intake was analyzed using 16S rRNA gene sequencing to assess microbial diversity, microbial composition at the order level, and the proportion of BAPB. Fecal butyric acid concentration was also measured. There were no significant differences in diversities and microbial composition between samples. The proportion of BAPB increased significantly after the intake (from 5.9% to 8.2%). However, one month after stopping the intake, the proportion of BAPB returned to the original value (5.4%). Fecal butyric acid concentration increased significantly from 0.99 mg/g feces before intake to 1.43 mg/g after intake (*p* = 0.028), which decreased significantly to 0.87 mg/g after stopping intake (*p* = 0.008). As BAPB produce butyric acid by degrading dietary fiber, functional barley may act as a prebiotic, increasing BAPB and consequently butyric acid in the intestine.

## 1. Introduction

More than 100 trillion bacteria, including 1000 different species, form the gut microbiota in the human intestine, greatly affecting human health [1]. Butyric acid, one of the short chain fatty acids (SCFA) produced by butyric acid-producing bacteria (BAPB) (e.g., portion of *Alistipes* and *Porphyromonas* (order Bacteroidales); *Clostridium*, *Eubacterium*, *Feacalibacterium*, and *Roseburia* (order Clostridiales)) in the intestine, has garnered attention owing to its role in the maturation of regulatory T cells in the intestine, which suppress excessive immune responses [2,3]. A decrease in BAPB in the intestine and in fecal butyric acid concentration has been reported in patients with idiopathic nephrotic syndrome [4,5], food allergy [6], cerebrovascular disease [7], and type 2 diabetes [8,9]. Therefore, increasing BAPB in the gut microbiota may aid in the prevention and treatment of various chronic diseases. Although probiotics have been employed extensively to correct gut microbiota imbalance (dysbiosis), their effects are limited, as most probiotics only pass through the intestine without residing in and modulating the gut microbial composition [10,11]. Instead, the efficacy of prebiotics has been receiving attention recently. According to the International Scientific Association for Probiotics and Prebiotics, prebiotics are defined as “a substrate that is selectively utilized by host microorganisms conferring a health benefit” and include dietary fibers such as β-glucan, fructan, resistant starch, and oligosaccharides [12]. Whole-grain products using corn, wheat, and barley are known to regulate the gut microbiota owing to their high content of dietary fiber [13,14,15]. BARLEYmax^®^, a line of non-genetically modified barley developed by the Commonwealth Scientific and Industrial Research Organization, contains twice the amount of dietary fiber and four times the resistant starch compared to regular barley and is reported to elevate the fecal SCFA concentration [16]. However, no study has analyzed its effect on the human gut microbiota, especially focusing on butyric acid. Therefore, we evaluated whether the intake of fiber-rich barley, BARLEYmax, elevates fecal butyric acid concentration by increasing BAPB in the gut of healthy adults.

## 2. Results

### 2.1. Participants’ Characteristics

A total of the 20 participants took part in this study; however, the analysis was performed on 18 participants after excluding two participants (owing to antibiotic use during the study period). The median age of the subjects, including 12 males (67%), was 35.9 years (IQR, 33.8–41.6 years). Seventeen participants (94%) were born vaginally. Median body mass index was 22.0 (19.6–24.6). Fifteen participants had allergic diseases, mostly allergic rhinitis (14 participants). Three participants were taking anti-histamine drugs for allergic rhinitis and one participant was taking magnesium oxide for constipation. Twelve participants (67%) were taking probiotics constantly. The compliance rate for BARLEYmax (the percentage of days for which participants had BARLEYmax during the 28-day study period) was 78% (64–93%) (Table 1).

### 2.2. Alpha Diversity

There was no significant difference in the numbers of observed species between the pre, post, and 1M groups (46 [IQR, 38–50], 45 [42–50], and 44 [38–50], respectively, *p* > 0.05). Moreover, there were no significant differences in Shannon and Simpson indices (Shannon index: 4.10 [3.96–4.41], 4.01 [3.81–4.14], and 4.15 [3.53–4.23], respectively, *p* > 0.05; Simpson index: 0.91 [0.88–0.93], 0.89 [0.87–0.92], and 0.91 [0.85–0.92], respectively, *p* > 0.05) (Figure 1A).

### 2.3. Beta Diversity

To evaluate the differences in gut microbiota between the pre, post, and one-month groups, a principal coordinates analysis plot of Bray–Curtis dissimilarity was generated, and the sample was characterized in two dimensions. No apparent clusters were observed before and after the intake of BARLEYmax (Figure 1B).

### 2.4. Taxonomic Composition

At the order level, Clostridiales and Bacteroidales were predominant among the samples. Although not statistically significant, the abundance of Clostridiales before BARLEYmax intake was 33.3%, which decreased to 30.4% after one month of intake (*p* = 0.30). The abundance of Bacteroidales increased from 16.1% before intake to 19.5% after intake (*p* = 0.51). Other bacterial orders did not show any significant changes before and after intake (Figure 1C).

### 2.5. Comparison of BAPB

The proportion of BAPB was 5.9% (2.4–6.8%) before intake, which increased significantly to 8.2% (3.7–10.8%) after the intake. However, one month after stopping the intake, the proportion decreased to the same level as that before the intake (5.4% [2.3–9.0%]) (Figure 2A).

### 2.6. Comparison of Fecal Organic Acid Concentration

Fecal organic acid concentrations were estimated in the samples containing detectable levels of corresponding organic acids (14, 10, and 15 samples in the pre, post, and 1M groups, respectively). Representative chromatograms of SCFAs at the three time points of one participant are shown in Appendix A. Butyric acid concentration increased significantly from 0.99 mg/g (0.74–1.04 mg/g) before intake to 1.43 mg/g (1.05–1.58 mg/g) after intake (*p* = 0.028), which decreased significantly to 0.87 mg/g after stopping the intake (0.62–1.06 mg/g; *p* = 0.008) (Figure 2B, left). Propionic acid concentration increased significantly from 1.16 mg/g (0.98–1.29 mg/g) before intake to 1.82 mg/g (1.55–2.22 mg/g) after intake (*p* = 0.018), which decreased significantly to 1.00 mg/g (0.85–1.49 mg/g; *p* = 0.021) (Figure 2B, center). Acetic acid concentration increased significantly from 2.79 mg/g (2.54–3.14 mg/g) before intake to 4.63 mg/g (3.38–4.74 mg/g) after intake (*p* = 0.018), which decreased significantly to 2.85 mg/g (1.97–3.19 mg/g; *p* = 0.016) (Figure 2B, right).

### 2.7. Post-Hoc Power Analysis

A post hoc power analysis was conducted to evaluate the outcome of the relative abundance of BAPB. The effect size of data was 0.430. When the two-sided type I error was set to 0.05, the power was 0.795, which was almost equivalent to the power 0.8 set as priori power analysis. Therefore, the initial sample size was confirmed to be efficient.

## 3. Discussion

In this study, we compared the gut microbiota of 18 healthy adults before and after BARLEYmax intake. We found that BARLEYmax increased the proportion of BAPB in the gut microbiota without a substantial change in the gut microbial composition. Moreover, we observed a corresponding increase in the concentration of fecal organic acids, including butyric acid. BARLEYmax is rich in dietary fibers such as β-glucan, fructan, and resistant starch. These components may have worked as prebiotics in the gut, resulting in the increase in BAPB.

It has been reported that functional barley intake increases the fecal SCFA concentration [16]. Bird et al. conducted a randomized crossover study in 17 healthy adult men consuming the Himalaya 292 grain (another line of fiber-rich barley), whole wheat, or refined cereal-based foods daily for 4 weeks [16]. The butyric, propionic, and acetic acids were significantly higher in samples collected 48 h after the intake of Himalaya 292 grain than in samples collected after the intake of whole wheat or refined cereal. Our study reports a similar increase in fecal organic acids following the intake of BARLEYmax. However, Bird et al. did not assess the changes in the gut microbiota and their correlation with the fecal organic acids. To our knowledge, this is the first study demonstrating that BARLEYmax increased the fecal organic acid concentration through the changes in the gut microbiota in humans.

Here, we discuss the components of BARLEYmax that may function as prebiotics, thus altering the gut microbiota and organic acid production. Aoe et al. randomly divided male Sprague Dawley rats into three groups that were fed BARLEYmax, BG012 (a high β-glucan barley line), or diet containing 5% cellulose (control) and analyzed the fecal SCFAs and gut microbiota in the cecal and distal colonic digesta. The concentration of total SCFAs was significantly high only in the BARLEYmax group [17]. This study suggested the possibility that fructan and resistant starch, rather than β-glucan, affects the gut microbiota and production of organic acids.

Although inulin, a type of fructan, was reported to affect the gut environment in several studies, it did not affect the butyric acid production in the gut. Neyrinck et al. reported that three months of 12 g/day inulin intake did not affect the fecal SCFA concentration in adults with obesity [18]. Similarly, a placebo-controlled, double-blind crossover trial on 50 healthy adults using 7 g of inulin or placebo per day reported that although the proportion of Bifidobacterium was higher in the inulin group, SCFA did not differ between the groups [19].

Resistant starch, relatively indigestible in the small intestine, is reported to reach the colon and change the gut microbiota. Walker et al. conducted a randomized crossover trial comparing the gut microbiota in 14 obese male volunteers consuming meals rich in either type-3 resistant starch or non-starch polysaccharides, for three weeks. The group that consumed resistant starch showed increased abundance of *Ruminococcus bromii* and *Eubacterium rectale* [20]. Similar results were reported in another study conducted in the participants consuming meals with type-2 resistant starch [21]. *R. bromii* and *E. rectale* are both BAPB, suggesting that resistant starch may promote BAPB growth in the gut.

In our study, BARLEYmax intake led to a moderate increase in Bacteroidales and a decrease in Clostridiales, though not significant. These results are strikingly similar to those obtained by Aoe et al., where the abundance of Bacteroidetes (includes the order Bacteriodales) was higher and that of Firmicutes (includes the order Clostridiales) was lower in the colonic digesta of rats fed BARLEYmax [17]. A possible reason for the low statistical significance in our study may be related to the amount of functional barley intake. While the intake of dietary fiber in the aforementioned study in rats consisted solely of functional barley, the dietary fiber content in our study food was 5.72 g per 40 g of granola, which is a quarter of the recommended dietary fiber intake in Japanese men (21 g per day) [22].

Our study has several limitations. First, we cannot deny the possibility that ingredients other than functional barley present in the study food affected the gut microbiota. A randomized placebo-controlled study would address this drawback. Second, our study does not reveal the effects of gut microbiota alteration on the immune system. Further studies are recommended, including the determination of Tregs proportion in peripheral blood lymphocytes. Third, since our study predominantly includes young males with high prevalence of allergic diseases (83%), it may be difficult to extrapolate our result to the healthy adult population in a generalized manner. However, a recent report suggests a high prevalence of allergic diseases (approximately 50%, predominantly allergic rhinitis) in the Japanese population [23]. Therefore, we believe that our study subjects are not distinct outliers, and in fact may be a rough representation of the Japanese population with respect to allergic diseases. Lastly, we recruited subjects only from one region of Japan; hence, our findings may not be applicable to other areas or ethnic groups. However, because the gut microbiota varies by race and nation, our results are useful only as a genuine microbiota profile of Japanese people.

Studies on BAPB and butyric acid in the gut have recently gained increased interest, as these are reported to be lower in patients with various chronic diseases. We previously reported the decrease in BAPB in pediatric patients with idiopathic nephrotic syndrome and chicken egg allergy [4,6]. It is therefore expected that the consumption of BARLEYmax could increase BAPB and butyric acid in the intestine, thereby preventing and treating various chronic diseases. In conclusion, the intake of dietary fiber-rich barley increases the fecal butyric acid concentration by promoting the growth of BAPB in the intestine.

## 4. Materials and Methods

### 4.1. Participants and Study Design

This study was conducted between June 2020 and December 2020 at the Kansai Medical University, Osaka, Japan. Twenty healthy adults (aged 18–65) were enrolled in the study. Information on the mode of delivery at birth, history of allergic diseases, antibiotic consumption in the last 3 months, and intake of probiotics were collected using a questionnaire. Intake of probiotics was defined as four or more times per week of consuming yogurt, cheese, miso (fermented soybean paste), pickles, probiotic drinks, or probiotic supplements. Participants received 40 g of granola containing BARLEYmax once each day for more than four times per week, for four weeks. The participants were not instructed on the schedule and the method of intake of BARLEYmax. Information on the consumption of BARLEYmax alone, with yogurt, with milk, or with soymilk, was collected using a questionnaire. Stool samples were collected thrice; at the start of eating granola, one month after eating granola, and one month after finishing eating granola (pre, post, and 1M, respectively). The participants were excluded from the study if they did not consume granola at least four times per week or consumed antibiotics during the study period. Stool samples were processed for 16S rRNA gene sequencing and the estimation of organic acids. The primary outcome was the changes in the relative abundance of BAPB. BAPB were defined as the 61 bacterial species reported in a previous study [24]. The secondary outcomes were the alpha and beta diversity, relative abundance of bacteria at the order level, and fecal organic acid concentration.

### 4.2. Study Food: BARLEYmax Granola

The study food included 40 g of BARLEYmax granola (TEIJIN LIMITED, Tokyo, Japan) containing 20.4 g of BARLEYmax. It consisted of brown sugar syrup, oats, puffed brown rice, coconut, rice oil, prune puree, dried mango, dried papaya, dried pineapple, and citric acid. Regarding the nutritional facts, 40 g of the study food contained 169 kilocalories, 3.32 g of protein, 5.76 g of fat, 23.08 g of sugar, 5.72 g of dietary fiber, and 0.48 g of resistant starch. Dietary fiber constituted 1.2 g of β-glucan and 2.32 g of fructan. The nutrient components were analyzed by the Japan Food Research Laboratories (Tokyo, Japan). The amount of total resistant starch, β-glucan, and fructan were measured using the resistant starch kit, mixed-linkage beta-glucan kit, and fructan kit, respectively (Megazyme, Sydney, Australia).

### 4.3. Stool Sampling and 16S rRNA Gene Sequencing

Stool samples (2 g) were collected per sampling by using a small sterilized spoon provided with the container and stored at −80 °C within 4 h of defecation until further analysis. The stool samples were thawed, and DNA was extracted using a Nucleo Spin DNA Stool Kit (MACHEREY-NAGEL, Düren, Germany). The seven hypervariable regions of the DNA, excluding v1 and v5 of the 16S rRNA region, were amplified using a 16S metagenomics kit following the manufacturer’s instructions (Thermo Fisher Scientific, Waltham, MA, USA). After refinement, a library was generated using the Ion Plus Fragment Library Kit and Ion Xpress Barcode Adapters Kit (Thermo Fisher Scientific, Waltham, MA, USA). The barcoded library was quantified and pooled to generate the final concentration of 50 pM per target using the Ion Universal Library Quantification Kit with the Quant Studio 5 system (Thermo Fisher Scientific, Waltham, MA, USA). The Ion Chef Instrument and the corresponding kit were used to achieve target concentrations for template preparation and emulsion PCR, and sequence analysis was performed using the Ion Gene Studio S5 System and Ion 530 chip (Thermo Fisher Scientific, Waltham, MA, USA). The sequence data were analyzed using Ion Reporter Software with the Metagenomics 16S w1.1 v5.16 workflow (Thermo Fisher Scientific, Waltham, MA, USA).

### 4.4. High Performance Liquid Chromatography

SCFA concentration was measured by TechnoSuruga Laboratory Co., Ltd., Shizuoka, Japan. Briefly, 0.1 g of feces was placed in a 2.0 mL tube with zirconia beads and suspended in MilliQ water. The samples were heated at 85 °C for 15 min, vortexed at 5 m/s for 45 s using FastPrep 24 5G (MP Biomedicals, CA, USA), and centrifuged at 15,350× *g* for 10 min. The filtered supernatants were measured for organic acids (acetic, propionic, and butyric acids) using high performance liquid chromatographic system (Prominence, SHIMADZU, Kyoto, Japan) consisting of a post-column reaction, a detector (CDD-10A, SHIMADZU, Kyoto, Japan), two tandemly arranged columns (Shim-pack SCR-102 (H), 300 mm × 8 mm ID, SHIMADZU, Kyoto, Japan), and a guard column (Shim-pack SCR-102 (H), 50 mm × 6 mm ID, SHIMADZU, Kyoto, Japan) [25]. The system was used with a mobile phase (5 mM p-toluenesulfonic acid) and a reaction solution (5 mM p-toluenesulfonic acid, 100 µM EDTA, and 20 mM Bis-Tris). The flow rate and oven temperature were 0.8 mL/min and 45 °C, respectively. The detector cell temperature was maintained at 48 °C.

### 4.5. Statistical Analysis and Sample Size

Continuous variables were expressed as medians and interquartile ranges (IQR), and categorical variables were expressed as numbers and percentages. The Wilcoxon signed-rank test was used for statistical analysis, and a *p*-value of less than 0.05 was considered statistically significant. To calculate the appropriate sample size for the study, we conducted a priori power analysis using G*Power version 3.1.9.4 (Heinrich-Heine University, Düsseldorf, Germany) [26], with an effect size of 0.8 and two-sided type I error of 0.05. Because a sample size of 15 would provide a power of 0.8, 20 people were recruited, owing to the dropout rate estimation of 25%. To verify the results, a post hoc power analysis for the primary outcome was performed with a two-sided type I error of 0.05.

## Figures and Tables

**Figure 1 metabolites-11-00559-f001:**
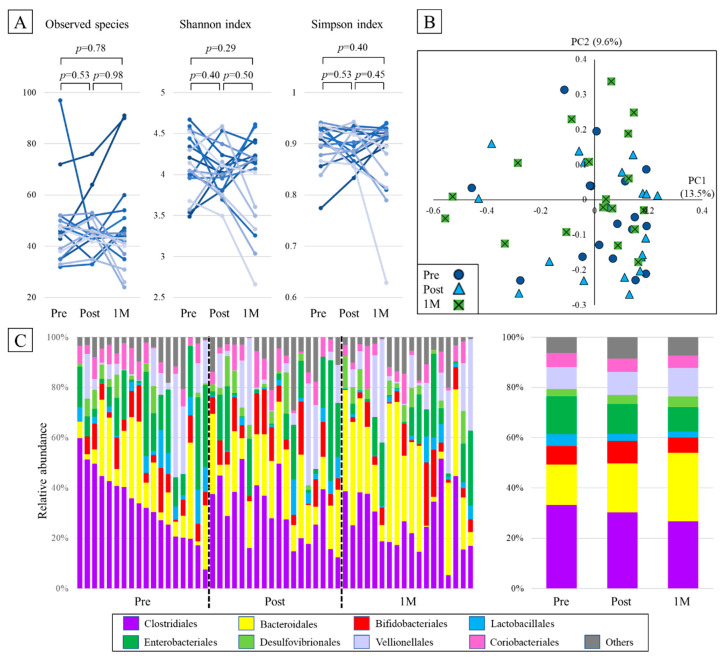
Changes in gut microbiota diversity and composition due to the intake of BARLEYmax: (**A**) numbers of species and Shannon and Simpson indices. (**B**) principal coordinates analysis plot of Bray–Curtis dissimilarity. Each point represents a sample. The circular, triangular, and square points represent the samples from the pre, post, and one-month groups, respectively. (**C**) Compositions of the gut microbiota at the order level. Each bar represents an individual sample (left) or a group (right).

**Figure 2 metabolites-11-00559-f002:**
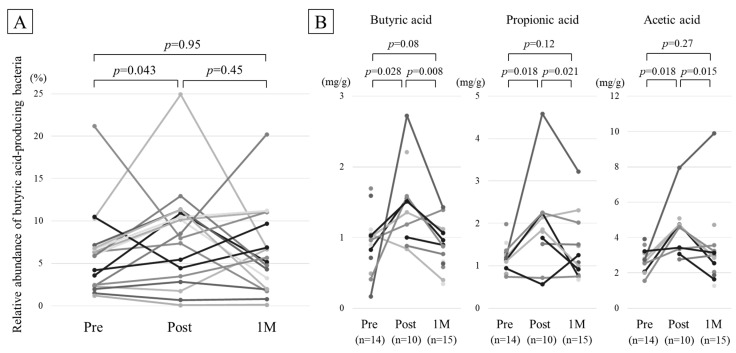
Changes in the composition of butyric acid-producing bacteria and fecal butyric acid concentration due to intake of functional barley. (**A**) Relative abundance of butyric acid-producing bacteria. (**B**) Concentrations of fecal butyric, propionic, and acetic acids. Points are not connected in the instances of insufficient data.

**Table 1 metabolites-11-00559-t001:** Characteristics of participants.

	n = 18
Sex, male (%)	12 (67%)
Age (years)	35.9 (33.8–41.6)
Body mass index	22.0 (19.6–24.6)
Mode of delivery, vaginal delivery (%)	17 (94%)
Allergic disease (%)	15 (83%)
Allergic rhinitis (%)	14 (78%)
Atopic dermatitis (%)	1 (6%)
Daily use of prescription drugs (%)	4 (22%)
Antihistamine (%)	3 (17%)
Magnesium oxide (%)	1 (6%)
Daily use of probiotics (%) ^†^	12 (66%)
Compliance rate (%) ^‡^	79% (64–93%)
Consumed BARLEYmax granola	
BARLEYmax alone	6 (33%)
With Milk	8 (44%)
With Yogurt	3 (17%)
With Soy milk	1 (6%)

Data are expressed as number (%) or median (IQR). ^†^ Four or more times per week of eating yogurt, cheese, miso (fermented soybean paste), pickles, probiotic drink, or probiotic supplements. ^‡^ Percentage of days for which participants took granola during the 28-day study period.

## Data Availability

The datasets used and analyzed during the current study can be found in the Kansai Medical University Research Data Storage and available from the corresponding author on reasonable request. The data are not publicly available due to privacy restrictions.

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
