# Peer review of "Fiber-Rich Barley Increases Butyric Acid-Producing Bacteria in the Human Gut Microbiota"

_metabolites, 2021, doi:10.3390/metabo11080559_

Round 1

Reviewer 1 Report

The paper submitted by Akagawa et al. is focused on the analysis of the gut microbiota diversity after the supplementation with fiber rich barley. The general impression on the manuscript is very good. It presents interesting data. In my opinion it should be published in Metabolites after addressing some minor point given below:

1) it is not clear for the reader how exactly BARLEYmax was given to participants - which form, how it was prepared for consumption; was it standardized in any way?

2) sample chromatogram for the quantification of organic acids in samples would be interesting for the reader

Author Response

Response to Reviewer #1:

The paper submitted by Akagawa et al. is focused on the analysis of the gut microbiota diversity after the supplementation with fiber rich barley. The general impression on the manuscript is very good. It presents interesting data. In my opinion it should be published in Metabolites after addressing some minor point given below:

Reply: We express our sincere gratitude for the positive and encouraging comments. We appreciate the time and effort you have invested in reviewing our manuscript.

1) it is not clear for the reader how exactly BARLEYmax was given to participants - which form, how it was prepared for consumption; was it standardized in any way?

Reply: Thank you for providing this insightful suggestion. The precise instructions of when and how to take BARLEYmax granola was not provided to the participants. We have added this point to the Materials and Methods section as follows: “The participants were not instructed on the schedule and the method of intake of BARLEYmax. Information on the consumption of BARLEYmax alone, with yogurt, with milk, or with soymilk, was collected using a questionnaire.” (Page 6, Line 210). Moreover, we have included the participant responses to the aforementioned questionnaire in Table 1.

2) sample chromatogram for the quantification of organic acids in samples would be interesting for the reader

Reply: Thank you for your suggestion. We totally agree with you. We have added the representative chromatograms of the three time points of one participant as Supplementary Figure 1.

Reviewer 2 Report

The submitted manuscript demonstrated that the intake of dietary fiber-rich barley increases the fecal butyric acid concentration by promoting the growth of butyric acid-producing bacteria in the intestine using 16S rRNA sequencing to assess microbial diversity, microbial composition. This topic is interesting and important in this field. The presented data is condensed and clear, and the article is well organized. In my opinion, this article is ready for publish. 

Author Response

Response to Reviewer #2:

The submitted manuscript demonstrated that the intake of dietary fiber-rich barley increases the fecal butyric acid concentration by promoting the growth of butyric acid-producing bacteria in the intestine using 16S rRNA sequencing to assess microbial diversity, microbial composition. This topic is interesting and important in this field. The presented data is condensed and clear, and the article is well organized. In my opinion, this article is ready for publish. 

Reply: We express our sincere gratitude for the positive and encouraging comments. We appreciate your time and effort invested in reviewing our manuscript.

Reviewer 3 Report

The authors investigated that the impact of fiber-rich barley on butyric acid-producing bacteria in human gut. The topic is interesting, and the experiment is relatively simple. However, there are some major concerns about the study design and research conclusion, which needs to be improved.

Major concerns:

  1. It has been reported that butyric acid-producing bacteria (BAPB)mainly belong to phylum Firmicutes, Order Clostridiales, species such as Eubacterium rectale and Faecalibacterium prausnitzii (DOI: 10.1111/j.1574-6968.2009.01514.x). In this study, the Order Clostridiales while the order Bacteroidales increased, the authors should indicate what kinds of gut microbiota are defined as BAPB in this study, and the orders of families of BAPB should be added in the introduction.
  2. Another major concern is that 83% of participants have an allergic disease, 22% of them take daily medicine. In addition, 66% of them take probiotics and 79% of participants took the granola during the study period. The average age is 35 (33.8-41.6), with 67% sex is male. All these conditions may impact the outcomes. The number of participates should be increased to make the conclusion shown in the title, especially adding the healthy individuals. Or modify the title to more specifically describe the results, such as by adding in adult human gut microbiota in patients with allergic disease.
  3. Since the study is relatively simple, the effect of an increase in BAPB in the human gut on disease (allergic diseases) or plasma inflammation markers, is suggested to be investigated to improve the quality of the manuscript. Even though the clinical investigation did not show probiotics can alter the composition of gut microbiota, but they have effects on the gut barrier independent of gut microbiota (DOI: 10.3390/ijms22136729) and immune regulation (DOI: 10.1016/j.clnesp.2021.06.020).

Minors:

  1. all the p values should be italicized, and all the P in lines 73-76 should be consistent with italic p.
  2. 16S rRNA sequencing should be 16S rRNA Ingene sequencing (line 17, same as in the method). The same change is needed in the Keywords (line 26).
  3. In Figure 2B, many dots are lacking connection. Is that because the lack of data at a time point or something else?

Author Response

Response to Reviewer #3:

The authors investigated that the impact of fiber-rich barley on butyric acid-producing bacteria in human gut. The topic is interesting, and the experiment is relatively simple. However, there are some major concerns about the study design and research conclusion, which needs to be improved.

Reply: We appreciate the time and effort you have invested in reviewing our manuscript. Your insightful comments with important references helped us significantly improve the manuscript.

Major concerns:

  1. It has been reported that butyric acid-producing bacteria (BAPB)mainly belong to phylum Firmicutes, Order Clostridiales, species such as Eubacterium rectale and Faecalibacterium prausnitzii (DOI: 10.1111/j.1574-6968.2009.01514.x). In this study, the Order Clostridiales while the order Bacteroidales increased, the authors should indicate what kinds of gut microbiota are defined as BAPB in this study, and the orders of families of BAPB should be added in the introduction.

Reply: Thank you for your important suggestion. Accordingly, we have added the genus and order information on well-known BAPB to the Introduction section as follows: “Butyric acid, one of the short chain fatty acids (SCFA) produced by butyric acid-producing bacteria (BAPB) (e.g., a portion of Alistipes and Porphyromonas (order Bacteroidales); Clostridium, Eubacterium, Feacalibacterium, and Roseburia (order Clostridiales)) in the intestine, has garnered attention owing to its role in the maturation of regulatory T cells in the intestine, which suppress excessive immune responses.” (Page 1, Line 31)

Also, the BAPB was defined more precisely in the Materials and Methods section as follows;

“BAPB was defined as the 61 bacterial species reported in a previous study [24]” (Page 6, Line 218)

  1. Another major concern is that 83% of participants have an allergic disease, 22% of them take daily medicine. In addition, 66% of them take probiotics and 79% of participants took the granola during the study period. The average age is 35 (33.8-41.6), with 67% sex is male. All these conditions may impact the outcomes. The number of participates should be increased to make the conclusion shown in the title, especially adding the healthy individuals. Or modify the title to more specifically describe the results, such as by adding in adult human gut microbiota in patients with allergic disease.

Reply: Thank you for your important comment. As you have pointed out, our study predominantly includes young men with allergic diseases. It may be difficult to generalize our result to the healthy adult population. Although we have considered limiting the study subjects to only allergic disease patients, the information on allergic diseases was based on self-report by patients, which lacks reliability. Moreover, according to a recent report on prevalence of allergic diseases (> 50%) in the Japanese population, we believe that our study subjects are not especially unique. We have added the above points to the limitations as follows:

“Third, since our study predominantly includes young males with high prevalence of allergic diseases (83%), it may be difficult to extrapolate our result to the healthy adult population in a generalized manner. However, a recent report suggests a high prevalence of allergic diseases (approximately 50%, predominantly allergic rhinitis) in the Japanese population [23]. Therefore, we believe that our study subjects are not distinct outliers, and in fact, may be a rough representation of the Japanese population with respect to allergic diseases.” (Page 6, Line 184)

  1. Since the study is relatively simple, the effect of an increase in BAPB in the human gut on disease (allergic diseases) or plasma inflammation markers, is suggested to be investigated to improve the quality of the manuscript. Even though the clinical investigation did not show probiotics can alter the composition of gut microbiota, but they have effects on the gut barrier independent of gut microbiota (DOI: 10.3390/ijms22136729) and immune regulation (DOI: 10.1016/j.clnesp.2021.06.020).

Reply: Thank you for highlighting this aspect. As you have pointed out, it is of great importance to investigate how the change in gut microbiota affects the immune system. However, unfortunately, we have not collected blood samples in this study. Therefore, we have added the following sentence as an important limitation:

“Second, our study does not reveal the effects of gut microbiota alteration on the immune system. Further studies are recommended, including the determination of Tregs proportion in peripheral blood lymphocytes.” (Page 6, Line 181)

Minors:

  1. all the p values should be italicized, and all the P in lines 73-76 should be consistent with italic p.

Reply: Thank you for your suggestion. All the P values in manuscript and figures have been italicized.

  1. 16S rRNA sequencing should be 16S rRNA Ingene sequencing (line 17, same as in the method). The same change is needed in the Keywords (line 26).

Reply: Thank you for your suggestion. As you have mentioned, because we are sequencing the 16S rRNA “gene”, we have changed “16S rRNA sequencing” to “16S rRNA gene sequencing” throughout the manuscript. However, we chose to use “gene sequencing” instead of “Ingene sequencing” since we could not find studies using the term “Ingene sequencing” in Pubmed. We apologize in case we misunderstood your suggestion.

  1. In Figure 2B, many dots are lacking connection. Is that because the lack of data at a time point or something else?

Reply: Thank you for your question. We apologize for the confusion. As you have pointed out, the lack of connection is due to lack of data. This point has been added to the legend of Figure 2 as follows; “Points are not connected in the instances of insufficient data.”

Round 2

Reviewer 3 Report

Thank the authors addressed all the questions and improved the quality of the manuscript, suggesting to be accepted. Congratulation.